# The Influence of the pH and Salinity of Water in Breeding Sites on the Occurrence and Community Composition of Immature Mosquitoes in the Green Belt of the City of São Paulo, Brazil

**DOI:** 10.3390/insects12090797

**Published:** 2021-09-05

**Authors:** Laura Cristina Multini, Rafael Oliveira-Christe, Antônio Ralph Medeiros-Sousa, Eduardo Evangelista, Karolina Morales Barrio-Nuevo, Luis Filipe Mucci, Walter Ceretti-Junior, Amanda Alves Camargo, André Barretto Bruno Wilke, Mauro Toledo Marrelli

**Affiliations:** 1Department of Epidemiology, School of Public Health, University of São Paulo, São Paulo 01246-904, SP, Brazil; aralphms@yahoo.com.br (A.R.M.-S.); karolinamorales182@gmail.com (K.M.B.-N.); cerettiw@usp.br (W.C.-J.); amanda.alca@uol.com.br (A.A.C.); 2Institute of Tropical Medicine, University of São Paulo, São Paulo 05403-000, SP, Brazil; christe.usp@gmail.com (R.O.-C.); entomol.edu@gmail.com (E.E.); 3Superintendency for the Control of Endemic Diseases, State Health Secretariat, São Paulo 12020-200, SP, Brazil; lfmucci@gmail.com; 4Department of Public Health Sciences, Miller School of Medicine, University of Miami, Miami, FL 33136, USA; axb1737@med.miami.edu

**Keywords:** arbovirus, atlantic forest, mosquito, malaria, ecology

## Abstract

**Simple Summary:**

Immature mosquitoes are found in natural and artificial aquatic habitats. Variations in physicochemical parameters of water, such as pH, salinity, conductivity, and total dissolved solids, in breeding habitats can influence larval occurrence and drive the proliferation of adult mosquitoes. Herein, we investigated the association between different values of physicochemical parameters in a variety of aquatic habitats and the occurrence and community composition of immature mosquito species in two environmentally protected areas in the city of São Paulo, Brazil. The aquatic habitats surveyed included epiphytic and ground bromeliads, bamboo internodes, ponds, tree hollows, lakes, and artificial containers. Our results revealed a statistically significant relationship between species occurrence and the variables of pH and salinity. The type of aquatic habitat also had a significant influence on mosquito species distribution. Investigating the interactions between immature mosquitoes and the environment in which they develop is important to elucidate the factors driving their occurrence and abundance, and could also be an important tool in planning and implementing immature mosquito control practices.

**Abstract:**

The physicochemical parameters of water, such as pH, salinity, conductivity, and total dissolved solids, can influence mosquito larval development, survival, and abundance. Therefore, it is important to elucidate how these factors influence mosquito occurrence. We hypothesized that the occurrence and community composition of immature mosquito species are driven not only by the availability of suitable aquatic habitats, but also by the physicochemical factors of these habitats. The primary objective of this study was therefore to investigate the influence of the physicochemical parameters of water in different types of aquatic habitats on the occurrence of mosquito species in two remnants of Atlantic Forest in the city of São Paulo, Brazil. Collections of immature mosquitoes and assessment of the physicochemical characteristics of the water in the collection sites were carried out for twelve months. The variation in species composition and occurrence with the measured physicochemical parameters and the type of breeding site was assessed using constrained ordination methods. The results indicate that there was a statistically significant difference in species composition as a function of the different types of aquatic habitats, and that pH had an influence on species occurrence even when the variance explained by the type of aquatic habitat was removed from the analysis. There was a statistically significant association between mosquito species occurrence and pH and salinity, and the former had a significant influence on the mosquito species collected regardless of the type of aquatic habitat, showing that the pH of the breeding site water is an important factor in driving mosquito population dynamics and species distribution.

## 1. Introduction

Mosquitoes are important vectors of diseases such as malaria, dengue (DENV), Zika (ZIKV), and chikungunya (CHIKV), which infect approximately 300 million people every year [1]. They are holometabolous insects (i.e., they undergo complete metamorphosis), and their immature forms inhabit exclusively aquatic habitats, whereas the adults are found in terrestrial habitats [2]. A wide range of aquatic habitats are explored by these insects, and their type and availability can drive mosquito species occurrence and abundance [3]. The aquatic habitats used by mosquitoes can be classified as natural (e.g., lakes, ponds, and bromeliads) or artificial (e.g., tires, cemetery urns, and plastic containers) [4], and as permanent, semi-permanent, or temporary [5]. In this context, the physicochemical properties of the water in breeding sites are an important factor for immature mosquito development and proliferation, and directly impact mosquito species composition [6].

Many biotic factors, such as feeding, predation, and intra- and inter-specific competition, and abiotic factors, including water salinity, pH, conductivity, and total dissolved solids, can influence immature mosquito development [7,8,9]. The physicochemical parameters of aquatic habitats can affect the biological fitness of mosquito larvae, larval growth and development, and the diversity and occurrence of immature mosquitoes [9,10,11]. It is therefore important to elucidate how these factors affect the occurrence of immature mosquitoes, which in turn influences the abundance of adult mosquitoes. Several habitat physicochemical variables have been shown to influence immature mosquito development and abundance [10,12,13], pH and salinity being considered the most important for mosquito occurrence [12].

Both pH and salinity are affected by complex interactions between natural and anthropogenic activities, highlighting the importance of having data on these parameters in vector breeding sites [14,15]. For aquatic organisms, pH is an important factor that limits their abundance and distribution as it is directly related to cell functioning and acts on the permeability of the cell membrane [16]. Mosquitoes are naturally tolerant to extreme pH values, which allows them to explore several types of environments [13,16,17], and some vector mosquito species have mechanisms that allow them to inhabit both acid and alkaline aquatic habitats, showing that pH tolerance can be associated with the abundance of these species [13,16].

Salinity (i.e., the concentration of mineral salts dissolved in the water) acts indirectly in the control of the metabolism of aquatic organisms and the productivity of ecosystems. In aquatic habitats, salinity tends to increase significantly in temporary waters, affecting the pH of the water and osmoregulatory mechanisms in insects [14]. Aquatic insects have important mechanisms for osmotic regulation that allow them to maintain the internal and external ionic concentration [18]. For example, although considered freshwater osmoregulators (i.e., they regulate the osmotic concentration of their hemolymph regardless of the concentration of the external medium), *Ae. aegypti* and *Aedes albopictus* larvae have been found in brackish water [18,19].

Understanding the effect of the physicochemical parameters of water in breeding sites on the occurrence of immature mosquitoes is important to elucidate the role these parameters play in limiting mosquito species richness, and to provide information on their potential use in the control of immature mosquitoes. We hypothesized that the occurrence and community composition of immature mosquito species are driven not only by the availability of suitable aquatic habitats, but also by the physicochemical parameters of these habitats. Therefore, the primary objective of this study was to investigate the association between different values of physicochemical parameters in a variety of aquatic habitats and the occurrence and community composition of immature mosquito species in two remnants of the Atlantic Forest in the city of São Paulo, Brazil.

## 2. Materials and Methods

### 2.1. Study Area

The study was conducted in two conservation areas composed of remnants of Atlantic Forest in the city of São Paulo, Brazil: the Capivari-Monos conservation area (APA) in the extreme south of the city and the Cantareira state park (PEC) in the extreme north of the city. Mosquito collections were carried out at seven sites in the two areas (Table 1).

### 2.2. Mosquito Collections and Species Identification

Mosquito collections were conducted monthly from November 2015 to February 2016, and from September 2016 to April 2017, totaling twelve months of collection. Immature mosquitoes (larvae and pupae) were collected in (1) artificial containers, (2) epiphytic bromeliads, (3) ground bromeliads, (4) bamboo stumps (the part of the bamboo left when the culm breaks or is broken transversely and the internode is exposed to rainwater), (5) bamboo with side holes caused by injury or phytophagous insects, (6) lakes, (7) tree hollows, (8) water accumulated in rocks, and (9) ponds. Several artificial aquatic habitats were explored, such as clay pots, plastic containers (cups, bottles, and buckets), bathtubs, animal drinking troughs, pet water bowls, cement boxes, water boxes, glass bottles, aluminum cans, swimming pools, tires, tanks, toilets, and pieces of bamboo that had been cut. All were considered artificial containers. The collections were carried out by actively seeking immature mosquitoes in aquatic habitats or by monitoring permanent aquatic habitats.

The water in small, naturally occurring aquatic habitats and large artificial containers was drained off with the aid of suction samplers (wash bottles coupled to a flexible hose); small artificial containers were completely emptied into basins, and larger breeding habitats, such as lakes, ponds, rocks and tanks, were surveyed with the aid of 60 or 500 mL ladle dippers. Immature mosquitoes were collected with pipettes, transported to the laboratory in 200 mL plastic pots filled with water from the breeding habitats, and reared in the laboratory until the emergence of adult mosquitoes, which were identified with the aid of taxonomic keys [2,4,20,21,22,23,24].

### 2.3. Physicochemical Parameters of the Water

The following physicochemical parameters of the water from aquatic habitats were measured in the field with a multiparameter waterproof meter (Hanna^®^ HI-9828, Hanna Instruments, Smithfield, RI, USA): pH, salinity (measured in psu-practical salinity unit), average conductivity (µS/cm), and total dissolved solids. Each sample from an aquatic habitat was transferred to a measuring cup and left for 10 min at room temperature to stabilize. The values of the parameters were recorded two minutes after the probe had been inserted and were added to a database for later correlation with the mosquito species identified.

### 2.4. Data Analysis

To achieve sampling sufficiency for each species, only those found in five or more breeding sites in which the physicochemical parameters of the water could be measured were considered in the statistical analysis. To assess whether the physicochemical variables measured were linearly correlated and to ensure that only explanatory non-collinear variables were included in the statistical models during the analyses, exploratory graphical analysis and Pearson’s correlation coefficient (r) analysis were performed. Constrained ordination methods were used to assess variations in species composition and the response to changes in the physicochemical factors and types of breeding sites. After transforming abundance data into occurrence or absence data, constrained analysis of proximities (CAP) was used with Sorensen’s similarity index [25] as the distance metric to investigate whether the type of aquatic habitat influences species composition. The model generated was tested with an ANOVA-like permutation test with 999 permutations to check for statistical significance. This test performs random permutations of the data and recalculates the model’s pseudo-F each time, generating a null distribution of pseudo-F values. Thus, the test calculates the probability of obtaining a value equal to or greater than the pseudo-F calculated in the original model by chance [26,27].

To investigate whether species occurrence is affected by physicochemical parameters, canonical correspondence analysis (CCA) was used. The model generated was tested with an ANOVA-like permutation test to check for statistical significance. A significance test was also performed for each variable to determine which of the measured physical and chemical factors had the greatest influence on species incidence. A partial CCA analysis was also run to control for all variation in mosquito incidence associated with differences in breeding site type.

To visualize the data, graphs were constructed with two different sets of information: the range of the values of the parameters for which each species is present, shown as boxplots without outliers, and the number of species identified for the corresponding value of the parameter.

All statistical and graphical analyses were performed in the R [28] computational environment with the vegan package [27].

## 3. Results

### 3.1. Immature Mosquito Species

A total of 10,650 immature mosquito specimens were collected, from which 9552 specimens from 16 genera comprising 73 species were correctly identified (Appendix A). Of these 73 species, 23 belonging to six genera occurred in five or more breeding sites and were considered in the statistical analyses. *Aedes fluviatilis* and *Limatus durhami* were found breeding predominantly in artificial aquatic habitats, whereas *Anopheles cruzii*, *Culex imitator, Culex worontzowi, Wyeomyia davisi* and *Wyeomyia theobaldi* were found breeding predominantly in epiphytic bromeliads (Figure 1).

Four medically important species were collected during the study: *Ae. aegypti*, *Ae. albopictus*, *An. Cruzii,* and *Haemagogus leucocelaenus*. Sixty-nine immature *Ae. aegypti* specimens were collected in 15 aquatic habitats, composed mainly of artificial containers and bamboo hollows (Appendix A). However, it was only possible to measure the physicochemical parameters of three of the fifteen aquatic habitats where *Ae. aegypti* was collected because of the reduced volume of water in the other twelve habitats. For this reason, this species was excluded from the statistical analysis.

A total of 143 immature *Ae. albopictus* specimens were collected in 36 aquatic habitats composed of artificial containers, bamboo hollows, and epiphytic bromeliads, and a total of 377 immature *An. cruzii* were collected in aquatic habitats in 138 different locations (Appendix A). This species is bromeliad-associated and was found only in epiphytic bromeliads and ground bromeliads. A total of 71 immature *Hg. leucocelaenus* were collected in 17 aquatic habitats composed of bamboo hollows, bamboo holes, and tree hollows (Appendix A). All the species except *An. cruzii* were found cohabiting the same aquatic habitats. *Aedes albopictus* and *Hg. leucocelaenus* were found in the same bamboo hollow. *Aedes aegypti* and *Ae. albopictus* were found in the same artificial container, and the other three species (*Hg. leucocelaenus*, *Ae. Albopictus,* and *Ae. aegypti*) were found in the same bamboo hollow.

### 3.2. Physicochemical Parameters

A total of 256 water samples were analyzed, among them were, artificial containers (21.87%), epiphytic bromeliads (45.31%), ground bromeliads (4.29%), bamboo holes (1.56%), lakes (12.89%), tree hollows (3.51%), bamboo hollows (3.12%), rock ponds (1.56%), and ponds (5.85%). In Cantareira state park (PEC), epiphytic bromeliads were the most prolific aquatic habitat for immature mosquitoes both in terms of the availability of this habitat and the number of mosquito specimens collected. Ground bromeliads were the least representative category in terms of availability, and tree hollows were the least representative aquatic habitat in terms of the number of specimens collected. In the Capivari-Monos environmental protection area (APA), the most prolific aquatic habitat for immature mosquitoes in terms of the number of plants available was epiphytic bromeliads, whereas the habitat in which the most specimens were collected was artificial containers. Bamboo hollows were the least representative aquatic habitat in terms of both their availability and the number of mosquito specimens collected.

In PEC, all the aquatic habitats had similar values of pH (3.7–8.86). Tree hollows varied more in conductivity (188–2633 µS/cm) and total dissolved solids (94–1316). Bamboo hollows had the highest salinities (0.09–1.37). In APA, artificial containers had the greatest pH variation (3.84–10.45). Bamboo hollows varied more in conductivity (499–6147 µS/cm) and total dissolved solids (250–3074) and had the highest salinities (0.24–3.31) (Appendix A).

### 3.3. Analysis by Parameter

Conductivity (µS/cm), total dissolved solids, and salinity were highly correlated in the collinearity analysis (r > 0.9). Because of the greater influence that salinity and pH have on aquatic organisms, these two variables were maintained in the statistical analyses, and conductivity and total dissolved solids were excluded. The model obtained from the CAP analysis for species occurrence or absence data using the Sorensen distance to calculate whether the type of aquatic habitat has an influence on species composition was statistically significant, indicating that there is a statistically significant difference in the composition of species between the different types of aquatic habitats (Table 2).

Canonical correspondence analysis (before the effect of the type of aquatic habitat was removed) revealed a statistically significant relationship between species occurrence and the variables pH and salinity (Table 2). Analysis of these variables separately in the CCA model showed that the relationship between species and pH (*pseudo-F* = 7.00; *p* = 0.015) better explains the dataset than the relationship between species and salinity (*pseudo-F =* 1.86; *p* = 0.322), showing that pH has a greater influence on species occurrence or absence than salinity.

The partial CCA to compare the occurrence of immature mosquito species in habitats with different pHs and salinities excluding the proportion of variance explained by the variable type of aquatic habitat yielded a *p*-value of 0.07 (Table 2). The isolated effects of pH (*pseudo-F* = 2.43; *p* = 0.002) and salinity (*pseudo-F* = 0.59; *p* = 0.86) were also assessed in the partial CCA. When the two parameters were analyzed together, the explained variance was marginally significant. However, when the analysis was conducted separately, it showed that pH influences species occurrence even when the variable type of aquatic habitat is excluded from the analysis.

### 3.4. Analysis by Mosquito Species

The analysis of the range of pHs within which each species was found and the number of species found for each pH showed that species occurrence varied greatly depending on habitat acidity (Figure 2). In the boxplot, there are two peaks of species occurrence, at pH ~5 and pH ~7, and as the pH becomes more acidic or basic, species occurrence decreases. However, some species had a high tolerance to acidic or basic mediums. *Aedes fluviatilis* and *Culex coronator*, for example, tolerated extreme values of pH, and specimens of these species were found in highly alkaline breeding sites (pH 10).

The analysis of the range of salinities within which each species was found and the number of species found for each salinity showed that the mosquito species observed here are found in aquatic habitats with lower salinities (Figure 3) and that species occurrence decreases as salinity increases. However, some *Culex* species, such as *Culex iridescens* and *Culex dolosus*, tolerated high salinities.

## 4. Discussion

Many factors can influence the distribution and community composition of immature mosquito species, including urbanization, climate, vegetation, interspecific association, and the physicochemical parameters of the water in the habitat [3,6]. This study investigated the association between different values of physicochemical parameters in a variety of aquatic habitats and the occurrence and community composition of immature mosquito species in São Paulo.

Our findings suggest that pH is the most important of the physicochemical parameters analyzed here, and that according to the results, it can influence species occurrence, regardless of the type of aquatic habitat, revealing that the quality of the water is an important factor in mosquito species occurrence. Even though our results show that different mosquito species are resistant to different values of pH, immature forms of some species were found in both acidic and alkaline aquatic habitats. Mosquito species occurrence was higher at pH ~7, indicating that neutral aquatic habitats represent an optimal medium for immature mosquitoes. Although the pH of the water in a breeding site is directly related to and can limit the distribution of aquatic organisms, some organisms are resistant to variations in and extreme values of pH [17]. As the medium becomes more acidic or more alkaline, some species need mechanisms that allow them to survive in higher or lower pHs, resulting in a decrease in species diversity and increased abundance of the few species that can survive in such conditions [29].

Our results also show that pH and salinity are important parameters driving the occurrence and distribution of immature mosquito species. When pH and salinity were analyzed together, there was a statistically significant association between these two variables and the occurrence of species, as well as the type of aquatic habitat. However, when analyzed separately, only pH was shown to have a positive association with species occurrence. In a previous study from our group, pH and salinity were the physicochemical factors most closely related to variations in mosquito species composition [12]. Their results also revealed that pH was an important predictor of *Ae. albopictus* and *Ae. aegypti* abundance [12].

*Aedes albopictus* has frequently been found in aquatic habitats with pHs ranging from acidic to alkaline, indicating that it has an efficient mechanism for regulating its body pH [17,29]. Species from the genus *Wyeomyia* have been found in bromeliads in the Amazon, where the pH varied from 4.9 to 6.9, and the correlation between pH and the occurrence of these species was statistically significant [13]. In the present study, *Wyeomyia* species were found in similar pH conditions and were associated with bromeliads, suggesting that the genus develops in more acidic environments.

Our results show that mosquito species in the study areas occurred more frequently in lower salinities. The species that had the highest occurrence in brackish aquatic habitats with different salinities were *Ae. albopictus*, *Cx. Iridescens,* and *Cx. dolosus*. Although *Ae. albopictus* larvae have previously been found in aquatic habitats with brackish water [19], under laboratory conditions the females of this species have higher rates of oviposition in mediums with lower salt concentrations, and egg productivity is also higher under these conditions [30], indicating that although *Ae. albopictus* preferentially inhabits mediums with lower salinity, it has a high tolerance to high salt concentrations.

Mosquito species develop different strategies to adapt to high levels of salinity, and tolerance to this factor varies between species [31]. Salinity and conductivity can be considered predictor variables for the occurrence of mosquito larvae, as an increase in the values of these parameters results in decreased species diversity, in turn increasing the abundance of salinity-tolerant species (e.g., *Ae. albopictus*) [6,29]. Several mosquito species vary greatly in their tolerance to salinity, and about 5% of mosquito species survive in very saline conditions [32].

Some *Culex* species are known to inhabit freshwater, while others preferentially inhabit saline water [31,32]. Among the species of this genus, *Cx. tarsalis* has a high salinity tolerance [33]. *Culex quinquefasciatus* has also shown high survival rates in saline water as this species is adapted to highly polluted aquatic habitats, and pollution increases salt concentrations in natural habitats [31]. *Culex dolosus* and *Cx. iridescens* are other species that seem to have developed mechanisms to tolerate high salinities, and are found in mediums with different salinities and in different types of aquatic habitats.

Our findings indicate that epiphytic bromeliads and artificial containers are important aquatic habitats in both study areas in terms of their number and the number of mosquito species and specimens that inhabit them. This may be because many of the artificial containers are permanent and their immature mosquito populations and the amount of water they contain are constant [34]. Likewise, as bromeliads tend to accumulate and retain water in their phytotelms, the occurrence of immature mosquitoes tends to remain constant in this habitat. The values for the coefficient of variation between the different aquatic habitats and the two study areas may be related to environmental factors such as precipitation, temperature, habitat type, size, and water volume, among other factors, that can affect physicochemical parameters and the species of immature mosquitoes found [14].

Bromeliads are important aquatic habitats for many organisms, including immature mosquitoes. In urban parks, they can serve as a refuge for several species of mosquitoes, including those of epidemiological importance [35], and they are excellent breeding sites for mosquitoes regardless of their location and the degree of anthropogenic changes [36]. Some groups of mosquitoes breed almost exclusively in bromeliads. Among these is the subgenus *Kerteszia* of the genus *Anopheles*, the subgenus *Microculex* of the genus *Culex,* and the species in the genus *Wyeomyia* [37,38,39]. Therefore, bromeliads are important aquatic habitats for mosquitoes and require monitoring, both in urban and rural environments [40].

*Culex* was the most representative group collected in the two study areas. While the majority of species from this genus tend to inhabit forested areas [2], they can also be found in great abundance in man-made habitats in suburban and urban areas [41,42]. We found a high number of artificial containers acting as aquatic habitats for the species collected, showing that the two areas are highly impacted by anthropogenic changes [43]. Although some collection sites are close to or in urban areas, artificial containers were found even in forested sites. In addition, the large number of *Cx. quinquefasciatus* collected in APA reveals the high degree of anthropogenic changes in this area, as this species is associated with the presence of humans. It should be kept in mind that artificial aquatic habitats can maintain some urban vector mosquito species in these areas.

*Aedes albopictus* colonizes a wide range of artificial and natural aquatic habitats and is considered a generalist in terms of its selection of oviposition sites, which can be in environments ranging from forested to highly urbanized areas [2,35,44]. The species is a competent vector for several arboviruses, including DENV, CHIKV, ZIKV, and yellow fever virus (YFV) [45,46,47,48,49]. In the present study, immature forms of *Ae. albopictus* were found in artificial containers and natural habitats, such as bromeliads and bamboos. As it can colonize several types of aquatic habitats in different types of environments, the species is an important bridge vector between wild and urban cycles of several mosquito-borne diseases, such as YFV [48,49,50,51].

Some species from the *Haemagogus* genus are responsible for the transmission of YFV in forested and rural areas in Brazil [52], and in several locations in the country, species in this genus are considered important vectors of YFV [53]. *Haemagogus leucocelanus* has been found inhabiting different types of environments, such as urban parks, forested areas close to urban areas [54], and forested areas close to dwellings [55,56]. In the present study, immature forms of this species were found in bamboos and tree hollows.

*Anopheles cruzii* is the primary vector of human and simian malaria in the Atlantic Forest [57]. A large number of *An. cruzii* specimens have been collected in the study areas in previous studies, highlighting the importance of this species in the bromeliad−malaria cycle in the Atlantic Forest, where it has been found in the vicinity of human dwellings [43,58,59,60]. The occurrence of this species, especially in PEC, raises concerns, as this conservation area harbors non-human primate populations that act as reservoirs for *Plasmodium*, increasing the likelihood of *Plasmodium* transmission to humans [57,61].

Understanding how the physicochemical parameters of aquatic habitats are responsible for driving the community composition of immature mosquitoes and why different species thrive under different conditions is essential to elucidate mosquito ecology and provide a better overview of the epidemiology of mosquito-borne disease transmission.

## 5. Conclusions

The present study revealed a significant association between mosquito species occurrence and habitat pH. The type of aquatic habitat also had a significant influence on mosquito species distribution. pH values had a statistically significant influence on mosquito species occurrence and community composition, regardless of the type of aquatic habitat, suggesting that this parameter drives mosquito population dynamics and species distribution. Investigating the interactions between immature mosquitoes and the environment in which they develop is important to elucidate the factors driving their incidence and abundance, and could also be an important tool in planning and implementing immature mosquito control practices.

## Figures and Tables

**Figure 1 insects-12-00797-f001:**
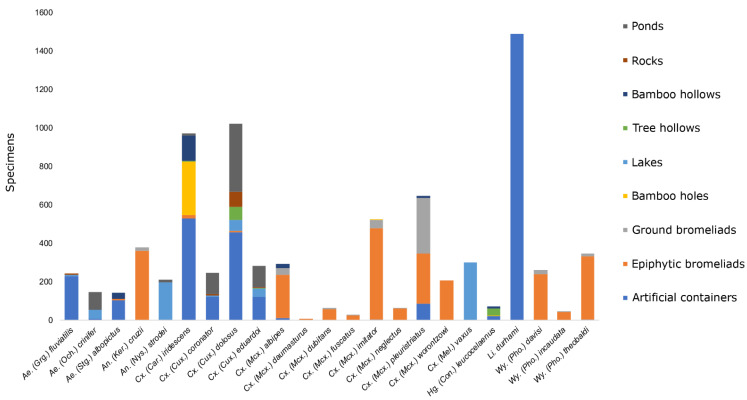
Community composition of immature mosquitoes collected in the aquatic habitats in two Atlantic Forest remnants in the city of São Paulo, Brazil, surveyed from February 2015 to April 2017.

**Figure 2 insects-12-00797-f002:**
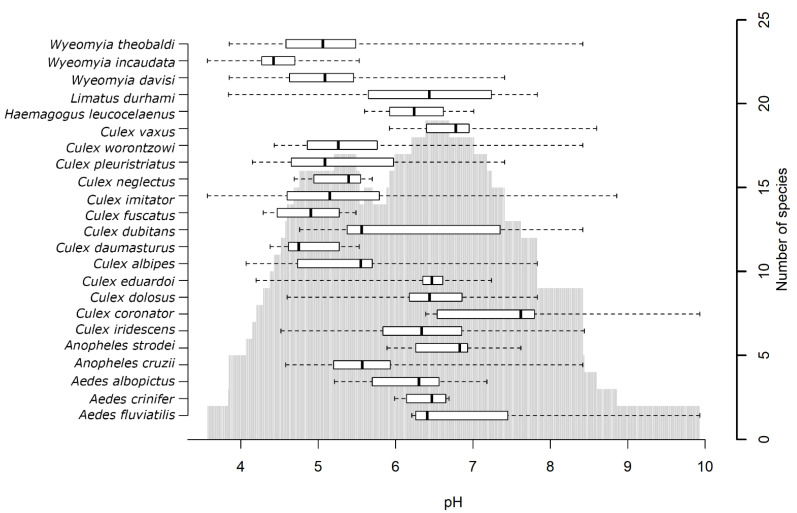
Variation in the number of species and specimens with pH of water from breeding sites. The boxplot (based on medians) and whiskers represent the range of pH values within which each species was found. The box represents the pH interval within which the species was observed most frequently. The vertical gray bars represent the number of species observed for the corresponding range of pHs.

**Figure 3 insects-12-00797-f003:**
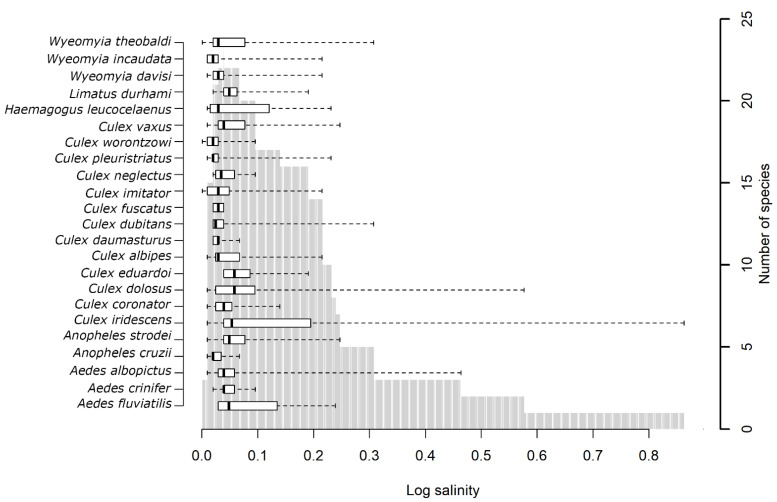
Variation in number of species and specimens with salinity (PSU on a log scale) of water from breeding sites. The boxplot (based on medians) and whiskers represent the range of values within which each species was found. The box represents the salinity interval within which the species was observed most frequently. The vertical gray bars represent the number of species observed for the corresponding range of salinities.

**Table 1 insects-12-00797-t001:** Location of collection sites and aquatic habitats in which immature mosquitoes were found.

Location	Collection Site	Geographical Coordinates
**PEC**	PEC-1	23°24′37.44” S 46°37′12.30” W
	PEC-2	23°26′51.90” S 46°38′5.46” W
	PEC-3	23°27′13.62” S 46°38′9.70” W
**APA**	APA-4	23°56′22.68” S 46°41′39.54” W
	APA-5	23°54′37.60” S 46°42′7.69” W
	APA-6	23°54′23.71” S 46°42′29.15” W
	APA-7	23°53′8.90” S 46°44′29.39” W

**Table 2 insects-12-00797-t002:** Results of the statistical analysis carried out to determine whether species occurrence is affected by physicochemical parameters and type of aquatic habitat.

Statistical Analysis					
**CAP**	Total inertia	Explained	Not explained	*p*	
	101.05	33.08	67.97	0.001	
**CCA**	Total inertia	Explained	Not explained	*p*	
	12.45	0.47	11.98	0.001	
**Partial CCA-excluding the effect of “type of aquatic habitat”**	Total inertia	Conditional	Explained	Not explained	*p*
	12.45	3.00	0.13	9.32	0.07

Total inertia = total variation of the data; explained (constrained inertia) = proportion of data explained by the predictive variables; not explained = proportion of data not explained by the predictive variables; *p* = the probability of obtaining a value of constrained inertia equal or greater than that observed—calculated with an ANOVA-like test with 999 permutations.

## Data Availability

All data are available in the article.

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
