# Peer review of "The Influence of the pH and Salinity of Water in Breeding Sites on the Occurrence and Community Composition of Immature Mosquitoes in the Green Belt of the City of São Paulo, Brazil"

_insects, 2021, doi:10.3390/insects12090797_

Round 1

Reviewer 1 Report

The study reports samples of mosquito larvae in aquatic sources, identify the mosquito larvae to species and measure aspects of the water such as pH, salinity and other physicochemical parameters. The manuscript is well written and describes the work in a clear manner. A few suggestions to make the paper better are:

In order to improve the paper and analysis more comparisons with the very similar study from the same group looking at species in parks in Sao Paulo. (Medeiros-Sousa, A.R.; de Oliveira-Christe, R.; Camargo, A.A.; Scinachi, C.A.; Milani, G.M.; Urbinatti, P.R.; Natal, D.; Ceretti-Junior, W.; Marrelli, M.T. Influence of water’s physical and chemical parameters on mosquito (Diptera: Culicidae) assemblages in larval habitats in urban parks of São Paulo, Brazil. Acta Trop. 2020, 205, 105394, doi:10.1016/j.actatropica.2020.105394. 45). Even though the two studies find many non-overlapping species a comparison of the species that do overlap would make it clearer for readers.

The sample sizes for some biotopes seem to be much smaller than others. They may be less frequent in the area. Even so, many specimens of each species were found. For example, several hundred Cx. iridescence in the four bamboo holes that were sampled. Were these all collected at one timepoint of during several trips?

By including the raw data in supplemental material it will be clearer how the experiments were carried out.

You write that 9535 larvae were collected. What proportion of mosquito larvae failed to hatch and could not be identified after collection? I assume that not all larvae hatched to adult mosquitoes that could be identified?

Author Response

Reviewer 1

The study reports samples of mosquito larvae in aquatic sources, identify the mosquito larvae to species and measure aspects of the water such as pH, salinity and other physicochemical parameters. The manuscript is well written and describes the work in a clear manner. A few suggestions to make the paper better are:

In order to improve the paper and analysis more comparisons with the very similar study from the same group looking at species in parks in Sao Paulo. (Medeiros-Sousa, A.R.; de Oliveira-Christe, R.; Camargo, A.A.; Scinachi, C.A.; Milani, G.M.; Urbinatti, P.R.; Natal, D.; Ceretti-Junior, W.; Marrelli, M.T. Influence of water’s physical and chemical parameters on mosquito (Diptera: Culicidae) assemblages in larval habitats in urban parks of São Paulo, Brazil. Acta Trop. 2020, 205, 105394, doi:10.1016/j.actatropica.2020.105394. 45). Even though the two studies find many non-overlapping species a comparison of the species that do overlap would make it clearer for readers.

Answer: We would like to thank the reviewer for the evaluation of our manuscript. We have made substantial changes to the manuscript, it was rewritten and reorganized. We strongly believe that we were able to fully address the points raised by the reviewer and therefore have improved significantly the overall quality of the manuscript.

We agree with the reviewer’s suggestion we have included a paragraph in the discussion section comparing the results of our present study with our previous study. Please, see below:

“Our results also show that pH and salinity are important parameters driving the occurrence and distribution of immature mosquito species. When pH and salinity were analyzed together, there was a statistically significant association between these two variables and the occurrence of species as well as the type of aquatic habitat. However, when analyzed separately, only pH was shown to have a positive association with species occurrence. In a previous study from our group, pH and salinity were the physicochemical factors most closely related to variations in mosquito species composition [12]. Their results also revealed that pH was an important predictor of Ae. albopictus and Ae. aegypti abundance [12].”

The sample sizes for some biotopes seem to be much smaller than others. They may be less frequent in the area. Even so, many specimens of each species were found. For example, several hundred Cx. iridescence in the four bamboo holes that were sampled. Were these all collected at one timepoint of during several trips?

Answer: We thank the reviewer for pointing out this inconsistency. Figure 1 shows the total number of immature mosquitoes collected by breeding sites in the study. Figure 1 shows several Cx. iridescence in bamboo holes because this species was collected several times in this type of habitat during several trips. However, only part of the breeding sites had their physicochemical parameters measured as many times there was not sufficient water to insert the probe, meaning that this species was found in more than four bamboo holes, but only for of this type of habitat had their physicochemical parameters measured.

By including the raw data in supplemental material it will be clearer how the experiments were carried out.

Answer: We thank the reviewer for the suggestion. We agree and have added a Supplementary table showing the number of collected species in both study areas. Please, see Table S1.

You write that 9535 larvae were collected. What proportion of mosquito larvae failed to hatch and could not be identified after collection? I assume that not all larvae hatched to adult mosquitoes that could be identified?

Answer: We thank the reviewer for pointing out this issue. All 9,535 immature mosquitoes were identified. We rephrased the sentence to improve clarity:

“A total of 10,650 immature mosquito specimens were collected, from which 9,535 specimens from 16 genera comprising 73 species were correctly identified.”

Reviewer 2 Report

Manuscript title: The influence of the pH and salinity of water in breeding sites  on the occurrence and community composition of immature mosquitoes in the green belt of the city of São Paulo, Brazil

The manuscript explains the association between pH, salinity of mosquito breeding sites on the occurrence and community composition of immature mosquitoes in Brazil in a convincing manner. Yet the efforts of research work done are not completely reflected in the manuscript. The whole manuscript shall be rewritten considering the following suggestions and avoiding formatting and typing errors, grammatical errors in the revised version to make it considerable for publication.

  1. Abstract – abstract shall be precise and concise, including a 2 line background and significant aspects of research followed by result and inference.
  2. Better sets of keywords shall be chosen.
  3. Introduction – the description of introduction does not follow a coherent order. Shall be rewritten following an orderly flow. Sentences with same meaning/concept are being repeated.
  4. Materials and methods - Aquatic habitats explored are repeated in table 1 and section 2.2
  5. How do the researchers delineate immaturity of the mosquito? based on what features..
  6. The rationality of physico-chemical parameters of water measured shall be given in introduction.
  7. Results – whole section shall be rewritten focusing only on the results obtained and not the inference. Any substantiations denoting to discuss the results shall be removed or moved to discussion section.
  8. Authors shall discuss why factors other than pH and salinity did not affect the mosquitoes. Discussion shall be rewritten depicting the obtained results and further substantiating with previous research work results, with appropriate citations. Was there any previous studies framed to apply a mosquito management plan based on analysing the mosquito breeding site physicochemical parameters?
  9. Conclusions?? or conclusion?

Author Response

Reviewer 2

The manuscript explains the association between pH, salinity of mosquito breeding sites on the occurrence and community composition of immature mosquitoes in Brazil in a convincing manner. Yet the efforts of research work done are not completely reflected in the manuscript. The whole manuscript shall be rewritten considering the following suggestions and avoiding formatting and typing errors, grammatical errors in the revised version to make it considerable for publication.

Answer: We would like to thank the reviewer for the evaluation of our manuscript. We have made substantial changes to the manuscript, it was rewritten and reorganized. We strongly believe that we were able to fully address the points raised by the reviewer and therefore have improved significantly the overall quality of the manuscript.

Abstract – abstract shall be precise and concise, including a 2 line background and significant aspects of research followed by result and inference.

Answer: We would like to thank the reviewer for the suggestion. We have rewritten and reorganized the abstract as recommended.

Better sets of keywords shall be chosen.

Answer: We thank the reviewer for the suggestion. However, the authors believe that the keywords provided are the most indicated to cover the scope of the manuscript.

Introduction – the description of introduction does not follow a coherent order. Shall be rewritten following an orderly flow. Sentences with same meaning/concept are being repeated.

Answer: Thank you for suggesting that improvements were needed in the introduction section. However, none of the reviewers made any suggestions deeming the introduction section adequate to provide enough background to position the readers on the subject being discussed in this manuscript. Furthermore, your suggestions were too broad not allowing us to completely grasp what were the points that needed improvement.

Materials and methods - Aquatic habitats explored are repeated in table 1 and section 2.2

Answer: We thank the reviewer for pointing out this issue. The aquatic habitats explored were excluded from table 1.

How do the researchers delineate immaturity of the mosquito? based on what features..

Answer: Basic mosquito biology dictates that mosquitoes are holometabolous insects that undergo complete metamorphosis. Mosquito immature forms, known as larvae and pupae inhabit exclusively aquatic habitats, while adults are found in terrestrial habitats. Mosquito larvae have four developmental instars, 1st, 2nd, 3rd, and 4th. Upon maturity, the 4th instar larvae molt into the pupal stage. The adults emerge directly from the pupal case on the surface of the water.

The rationality of physico-chemical parameters of water measured shall be given in introduction.

Answer: Thank you for your suggestion. We agree with the importance of having a clear rationale in the introduction section to support the hypothesis and objectives of our manuscript. We have provided the rationale of how physicochemical parameters of water affect mosquito development and proliferation in the last paragraph of the introduction section, please see it below:

“Understanding the effect of the physicochemical parameters of water in breeding sites on the occurrence of immature mosquitoes is important to elucidate the role these parameters play in limiting mosquito species richness and to provide information on their potential use in the control of immature mosquitoes. We hypothesized that the occurrence and community composition of immature mosquito species are driven not only by the availability of suitable aquatic habitats but also by the physicochemical parameters of these habitats. Therefore, the primary objective of this study was to investigate the association between different values of physicochemical parameters in a variety of aquatic habitats and the occurrence and community composition of immature mosquito species in two remnants of the Atlantic Forest in the city of São Paulo, Brazil.”

Results – whole section shall be rewritten focusing only on the results obtained and not the inference. Any substantiations denoting to discuss the results shall be removed or moved to discussion section.

Answer: We thank the reviewer for pointing out this issue. Inferences discussing the results were rephrased or excluded from the results section.

Authors shall discuss why factors other than pH and salinity did not affect the mosquitoes. Discussion shall be rewritten depicting the obtained results and further substantiating with previous research work results, with appropriate citations. Was there any previous studies framed to apply a mosquito management plan based on analysing the mosquito breeding site physicochemical parameters?

Answer: We thank the reviewer for the suggestion. We agree with the importance of discussing factors other than pH and salinity in the proliferation of mosquitoes. However, please keep in mind that our study was not designed to determine the mechanisms involved in the interaction between a given physicochemical parameter and mosquito development. Our objective in this manuscript was to investigate the association between different values of physicochemical parameters in a variety of aquatic habitats and the occurrence and community composition of immature mosquito species. Therefore, discussing how physicochemical parameters affected the mosquitoes would be speculation and would not fall into the scope of this manuscript. This study can serve as a stepping stone to future studies focusing on the physiological mechanisms used by different mosquito species to cope with the conditions found in different aquatic habitats.

Conclusions?? or conclusion?

Answer: We thank the reviewer for pointing out this issue. We changed the name of the section to Conclusions.

Reviewer 3 Report

Article “The influence of the pH and salinity of water in breeding sites on the occurrence and community composition of immature mosquitoes in the green belt of the city of São Paulo, Brazil” written by Multini et al.

In this manuscript, the authors aim to investigate whether the physiochemical parameters and the type of aquatic habitats affect the community composition of immature mosquito species. They found that physiochemical parameters and habitat type affect mosquitoes and that pH had a significant influence on the mosquito species, regardless of the type of habitat. The manuscript overall is well written and the methods are clearly described. Below my specific comments.

Line 47: I suggest to remove this sentence or “appease” it. Here, the authors analysed the role of few variables and some important parameters affecting mosquitoes, such as temperature, are not included. Therefore, other variables (also biotic) could act simultaneously to pH in influencing mosquito distribution and population dynamics.

Line 61: In this sentence, the authors mention the important role of biotic factors on mosquito development, but no references are included. They should add bibliography about that.

Line 200: delete space.

Line 293: put “and” before “mosquitoes”.

Line 349: put Anopheles in italic.

Line 367: put the species name in italic.

Line 370: delete space.

Line 383-384: Put Plasmodium in italic.

Table S1: What does CV means? Explaine it in caption.

Author Response

Reviewer 3

Article “The influence of the pH and salinity of water in breeding sites on the occurrence and community composition of immature mosquitoes in the green belt of the city of São Paulo, Brazil” written by Multini et al.

In this manuscript, the authors aim to investigate whether the physiochemical parameters and the type of aquatic habitats affect the community composition of immature mosquito species. They found that physiochemical parameters and habitat type affect mosquitoes and that pH had a significant influence on the mosquito species, regardless of the type of habitat. The manuscript overall is well written and the methods are clearly described. Below my specific comments.

Answer: We would like to thank the reviewer for the evaluation of our manuscript. We have made substantial changes to the manuscript, it was rewritten and reorganized. We strongly believe that we were able to fully address the points raised by the reviewer and therefore have improved significantly the overall quality of the manuscript.

Line 47: I suggest to remove this sentence or “appease” it. Here, the authors analysed the role of few variables and some important parameters affecting mosquitoes, such as temperature, are not included. Therefore, other variables (also biotic) could act simultaneously to pH in influencing mosquito distribution and population dynamics.

Answer: We would like to thank the reviewer for pointing out this issue. We agree with the reviewer and rephrased the sentence to improve clarity.

“There was a statistically significant association between mosquito species occurrence and pH and salinity, and the former had a significant influence on the mosquito species collected regardless of the type of aquatic habitat, showing that the pH of the breeding site water is an important factor in driving mosquito population dynamics and species distribution.”

Line 61: In this sentence, the authors mention the important role of biotic factors on mosquito development, but no references are included. They should add bibliography about that.

Answer: We thank the reviewer for the suggestion. We had already included a reference about biotic factors on mosquito development.

Please, see:

Dickson, L.B.; Jiolle, D.; Minard, G.; Moltini-Conclois, I.; Volant, S.; Ghozlane, A.; Bouchier, C.; Ayala, D.; Paupy, C.; Moro, C.V.; et al. Carryover effects of larval exposure to different environmental bacteria drive adult trait variation in a mosquito vector. Sci. Adv. 2017, 3, 1–15, doi:10.1126/sciadv.1700585.

Line 200: delete space.

            Answer: Thank you for the suggestion. The space has been deleted.

Line 293: put “and” before “mosquitoes”.

            Answer: Thank you for the suggestion.

Line 349: put Anopheles in italic.

            Answer: Thank you for pointing out this issue. The term has been corrected.

Line 367: put the species name in italic.

            Answer: Thank you for pointing out this issue. The term has been corrected.

Line 370: delete space.

            Answer: Thank you for pointing out this issue. The space has been deleted.

Line 383-384: Put Plasmodium in italic.

            Answer: Thank you for pointing out this issue. The term has been corrected.

Table S1: What does CV means? Explaine it in caption.

Answer: Thank you for pointing out this issue. CV means Coefficient of variation. We have added the meaning of ‘CV’ at the bottom of the supplementary table. Please, see Table S2.

Reviewer 4 Report

General comments: The manuscript dealing with the influence of pH and salinity of water in breeding sites on the occurrence and community composition of immature mosquitoes in the green belt of the city of Sao Paulo, Brazil contains valuable data on the occurrence of more than 20 mosquito species in the study area and their relationship to the pH and salinity of the water. However, in the present form the manuscript is not suitable for publication. On the one hand the English has to be improved, but more important is the improvement of the data presentation and accuracy of the data handling e.g. line 169.. a total of 9,535 immature mosquito specimens from 16 genera comprising 73 species were collected…. I see only 6 genera and 23 species. In the text Cx. quinquefasciatus is mentioned collected in APA (line 359) but not in the list of species.

The conclusions which are drawn are to general and are sometimes confusing e.g. ….our findings suggest that the pH is the most important parameter and it can limit species occurrence regardless of the type of aquatic habitat…. This statement is wrong, the type of habitat and the adjustment of the species biology to certain habitats (e.g. temporary or permanent) plays a more important role than the pH. However, it is right that species have a special preference for the quality of the water, such as the pH. The results on the effect of the pH on the occurrence of some species is well documented in Fig. 2. Figure 1 is hard to understand maybe a table would be more self-explanatory. I propose that the authors re-write the manuscript and refer more to the certain ecological situation in the study area and the occurrence of the mosquito species in the various habitat they have investigated but not in a general way as it is done in line 197-200,  Better to characterize the different habitats (e.g. as in line 57)  in a general way and refer to species which inhabit the various habitats.

Special remarks

Line 18…exclusively in natural and artificial aquatic habitats…. Of course this covers every type of breeding site.

Line 29…occurrence instead of incidence would be better.

Line 52….holometabolous.. has not be explained in a scientific paper.

Line 290… The higher of the two peaks….mosquitoes….. rephrase this sentence

Line 294….not only mosquitoes are resistant to extreme values of pH!

Line 310….Our results….salinity. This statement is too general. There are a lot of halophile mosquito species known… Ae. mariae, Ae. caspius etc.

Line 319-320, What is the meaning of this sentence?

Line 338 ..tanks… better phytotelms…

Line 349…Anopheles italic.
line 353-355…. Culex can be found also in great abundance in man-made habitats e.g. Cx. quiquefasciatus.

Line 367….Ae. albopictus italic.

Author Response

Reviewer 4

General comments: The manuscript dealing with the influence of pH and salinity of water in breeding sites on the occurrence and community composition of immature mosquitoes in the green belt of the city of Sao Paulo, Brazil contains valuable data on the occurrence of more than 20 mosquito species in the study area and their relationship to the pH and salinity of the water. However, in the present form the manuscript is not suitable for publication.

On the one hand the English has to be improved, but more important is the improvement of the data presentation and accuracy of the data handling e.g. line 169.. a total of 9,535 immature mosquito specimens from 16 genera comprising 73 species were collected…. I see only 6 genera and 23 species.

Answer: We would like to thank the reviewer for the evaluation of our manuscript. We have made substantial changes to the manuscript, it was rewritten and reorganized. We strongly believe that we were able to fully address the points raised by the reviewer and therefore have improved significantly the overall quality of the manuscript.

The manuscript has been professionally copyedited. We have added an explanation in the Results section as to why 23 species belonging to six genera have been added to the data analysis. Please, see below:

“A total of 10,650 immature mosquito specimens were collected, from which 9,535 specimens from 16 genera comprising 73 species were correctly identified. Of these 73 species, 23 belonging to six genera occurred in five or more breeding sites and were considered in the statistical analyses.”

In the text Cx. quinquefasciatus is mentioned collected in APA (line 359) but not in the list of species.

Answer: We thank the reviewer for pointing out this inconsistency. We have added a supplementary table showing the number of all species collected in both study areas. Please, see Table S1.

The conclusions which are drawn are to general and are sometimes confusing e.g. ….our findings suggest that the pH is the most important parameter and it can limit species occurrence regardless of the type of aquatic habitat…. This statement is wrong, the type of habitat and the adjustment of the species biology to certain habitats (e.g. temporary or permanent) plays a more important role than the pH. However, it is right that species have a special preference for the quality of the water, such as the pH.

Answer: We thank the reviewer for pointing out this issue. The sentence was rephrased to improve quality. Please, see below:

“Our findings suggest that pH is the most important of the physicochemical parameters analyzed here and that according to the results, it can influence species occurrence regardless of the type of aquatic habitat. Revealing that the quality of the water is an important factor on mosquito species occurrence.”

The results on the effect of the pH on the occurrence of some species is well documented in Fig. 2. Figure 1 is hard to understand maybe a table would be more self-explanatory. I propose that the authors re-write the manuscript and refer more to the certain ecological situation in the study area and the occurrence of the mosquito species in the various habitat they have investigated but not in a general way as it is done in line 197-200,  Better to characterize the different habitats (e.g. as in line 57)  in a general way and refer to species which inhabit the various habitats.

Answer: We thank the reviewer for pointing out these issues. We included a supplementary table showing the number of all species collected in the study by breeding sites to improve clarity. Please, see Table S1. We have also rewritten lines 197-200 to improve clarity. Please, see below:

“A total of 256 water samples were analyzed, among them were, artificial containers (21.87%), epiphytic bromeliads (45.31%), ground bromeliads (4.29%), bamboo holes (1.56%), lakes (12.89%), tree hollows (3.51%), bamboo hollows (3.12%),  rock ponds (1.56%) and ponds (5.85%).”

We agree with the reviewer with the importance of referring more to the certain ecological situation in the study area and the occurrence of the mosquito species in the various habitat. However, please keep in mind that our study was not designed to investigate what species inhabit what habitats. Our objective in this manuscript was to investigate the association between different values of physicochemical parameters in a variety of aquatic habitats and the occurrence and community composition of immature mosquito species regardless of what was the habitat. Therefore, reorganizing the text as suggested would not fall into the scope of this manuscript. This study can serve as a stepping stone to future studies focusing on specific aquatic habitats and ecosystems.

Special remarks

Line 18…exclusively in natural and artificial aquatic habitats…. Of course this covers every type of breeding site.

Answer: We thank the reviewer for pointing out this issue. The word “exclusively” has been deleted.

Line 29…occurrence instead of incidence would be better.

Answer: Thank you for the suggestion. The term ‘incidence’ has been replaced for ‘occurrence’ as suggested.

Line 52….holometabolous.. has not be explained in a scientific paper.

Answer: Thank you for pointing that out. Even though we agree, we would like to keep a short explanation of the holometabolous meaning to better position readers that are not familiar with the term.

Line 290… The higher of the two peaks….mosquitoes….. rephrase this sentence

Answer: We thank the reviewer for pointing out this issue. The sentence has been rephrased. Please, see below:

“Mosquito species occurrence was higher at pH ~7, indicating that neutral aquatic habitats represent an optimal medium for immature mosquitoes.”

Line 294….not only mosquitoes are resistant to extreme values of pH!

Answer: Thank you for pointing out this issue. The sentence has been rephrased. Please, see below:

“Although the pH of the water in a breeding site is directly related to and can limit the distribution of aquatic organisms, some organisms are resistant to variations in and extreme values of pH”.

Line 310….Our results….salinity. This statement is too general. There are a lot of halophile mosquito species known… Ae. mariae, Ae. caspius etc.

Answer: We thank the reviewer for pointing out this issue. The sentence has been rephrased as suggested. Please, see below:

“Our results also show that mosquito species in the study areas occurred more frequently in lower salinities.”

Line 319-320, What is the meaning of this sentence?

Answer: Thank you for pointing out the need to further explain the meaning of the sentence. Our original idea was to highlight how different levels of tolerance to salinity influenced the development and proliferation of vector mosquito species. We have modified the sentence to improve clarity as requested.

“Mosquito species develop different strategies to adapt to high levels of salinity, and tolerance to this factor varies between species.”

Line 338 ..tanks… better phytotelms…

Answer: Thank you for the suggestion. The term has been replaced as requested.

Line 349…Anopheles italic.

            Answer: Thank you for pointing out this issue. The term has been corrected.

line 353-355…. Culex can be found also in great abundance in man-made habitats e.g. Cx. quiquefasciatus.

Answer: Thank you for pointing out this issue. The sentence has been rephrased to improve clarity. Please, see below:

“While the majority of species from this genus tend to inhabit forested areas [2], they can also be found in great abundance in man-made habitats in suburban and urban areas [41,42].”

Line 367….Ae. albopictus italic.

Answer: Thank you for pointing out this issue. The term has been corrected.

Round 2

Reviewer 2 Report

Author did a good job in revision 

Author Response

Dear Reviewer

Thank you very much for reviewing our manuscript and for the suggestions, which have enabled us to greatly improve the text.

Sincerely,

Mauro Marrelli

Reviewer 4 Report

General comments: Most of the comments and recommendations were considered in the revised manuscript. However, I still cannot find Cx. quinquefasciatus in the species list, it is confusing, most probably this species was not found in at least 5 breeding sites. In my point of view the value of the paper would be increased when all 73 species which were precisely determined belonging to 16 genera would be mentioned in a list. In this list all species should be listed even if they are found in less than 5 breeding sites.

Author Response

Dear Reviewer

Thank you very much for reviewing our manuscript and for the suggestions, which have enabled us to greatly improve the text.

The supplementary S1 Table shows all mosquito species collected by breeding site in the two remnants of Atlantic Forest in the city of São Paulo, Brazil. Culex quinquefasciatus was only found  in Capivari-Monos conservation area (second area in the S1 Table)

Sincerely,

Mauro Marrelli